# The Genotoxic and Pro-Apoptotic Activities of Advanced Glycation End-Products (MAGE) Measured with Micronuclei Assay Are Inhibited by Their Low Molecular Mass Counterparts

**DOI:** 10.3390/genes12050729

**Published:** 2021-05-13

**Authors:** Monika Czech, Maria Konopacka, Jacek Rogoliński, Zbigniew Maniakowski, Magdalena Staniszewska, Łukasz Łaczmański, Danuta Witkowska, Andrzej Gamian

**Affiliations:** 1Dr. Józef Rostek Regional Hospital, Gamowska 3, 47-400 Racibórz, Poland; monnika.cz@gmail.com; 2Center for Translational Research and Molecular Biology of Cancer, Maria Skłodowska-Curie National Research Institute of Oncology Gliwice Branch, Wybrzeże Armii Krajowej 15, 44-100 Gliwice, Poland; marykonopacka@gmail.com (M.K.); jacek.rogolinski@io.gliwice.pl (J.R.); 3Department of Medical Physics, Maria Skłodowska-Curie National Research Institute of Oncology Gliwice Branch, Wybrzeże Armii Krajowej 15, 44-100 Gliwice, Poland; rogolinski@gmail.com; 4Laboratory of Separation and Spectroscopic Method Applications, Centre for Interdisciplinary Research, Faculty of Natural Sciences and Health, The John Paul II Catholic University of Lublin, Konstantynow 1J, 20-708 Lublin, Poland; magdalena.staniszewska@kul.pl; 5Hirszfeld Institute of Immunology and Experimental Therapy, Polish Academy of Sciences, Rudolfa Weigla 12, 53-114 Wrocław, Poland; lukasz.laczmanski@hirszfeld.pl (Ł.Ł.); danuta.witkowska@hirszfeld.pl (D.W.); 6Wrocław Research Center EIT+, PORT, Stabłowicka 147/149, 54-066 Wrocław, Poland

**Keywords:** advanced glycation, glycation products, advanced glycation end-products, MAGEs, genotoxicity, apoptosis, micronuclei assay

## Abstract

An association between the cancer invasive activities of cells and their exposure to advanced glycation end-products (AGEs) was described early in some reports. An incubation of cells with BSA–AGE (bovine serum albumin–AGE), BSA–carboxymethyllysine and BSA–methylglyoxal (BSA–MG) resulted in a significant increase in DNA damage. We examined the genotoxic activity of new products synthesized under nonaqueous conditions. These were high molecular mass MAGEs (HMW–MAGEs) formed from protein and melibiose and low molecular mass MAGEs (LMW–MAGEs) obtained from the melibiose and N-α-acetyllysine and N-α-acetylarginine. We have observed by measuring of micronuclei in human lymphocytes in vitro that the studied HMW–MAGEs expressed the genotoxicity. The number of micronuclei (MN) in lymphocytes reached 40.22 ± 5.34 promille (MN/1000CBL), compared to 28.80 ± 6.50 MN/1000 CBL for the reference BSA–MG, whereas a control value was 20.66 ± 1.39 MN/1000CBL. However, the LMW–MAGE fractions did not induce micronuclei formation in the culture of lymphocytes and partially protected DNA against damage in the cells irradiated with X-ray. Human melanoma and all other studied cells, such as bronchial epithelial cells, lung cancer cells and colorectal cancer cells, are susceptible to the genotoxic effects of HMW–MAGEs. The LMW–MAGEs are not genotoxic, while they inhibit HMW–MAGE genotoxic activity. With regard to apoptosis, it is induced with the HMW–MAGE compounds, in the p53 independent way, whereas the low molecular mass product inhibits the apoptosis induction. Further investigations will potentially indicate beneficial apoptotic effect on cancer cells.

## 1. Introduction

Advanced glycation end-product (AGE) formation and accumulation in tissues play an important role in diseases related to diabetes and ageing processes [1,2]. The glycation products are derived from condensation of carbonyl groups, mainly by reducing sugars, or low molecular mass α-oxoaldehydes with free amino groups of proteins, or amino acids. The first step of glycation is the formation of a reversible Schiff base [3,4]. This unstable bond undergoes an Amadori rearrangement, creating a more stable, but still partially reversible, Amadori product. This 1-amino-1-deoxyketose with carbonyl function is considered an early product of glycation, a precursor of further compounds formed in the process of glycation. Then, the complex cascade of dehydration, fragmentation, condensation, oxidation and cyclization reactions occurs. These reactions yield a variety of largely undefined advanced Maillard reaction end-products, so-called AGEs. In the presence of oxygen, the Amadori products are degraded to form even more reactive dicarbonyl derivatives; therefore, the term glycoxidation is also used for these reactions [5]. Among great number of possible structures, there are only few known AGEs with determined structural formula, including pentosidine, pyrraline, carboxymethyllysine (CML), derivatives of imidazole, argpyrimidine [6], cypentodine, arginine–lysine imidazole, imidazolone and glycolaldehyde–pyridine (GA–pyridine) [4,7,8]. The protein AGEs are relatively resistant to proteases, and part of the cross-linked protein material forms insoluble aggregates [9]. The free endogenous and exogenous (taken with food) AGEs may bind to their specific receptors, RAGE, and stimulate cellular response [10]. The AGE–RAGE interactions contribute to and are responsible for induction of oxidative stress, expression of adhesion molecules and activation of apoptosis [10,11,12]. Previously, an association between the cancer invasive activities of cells and their exposition to AGEs has been observed in melanoma and colorectal carcinoma cells [13,14]. Incubation of pig kidney cells with BSA–AGEs such as BSA–carboxymethyllysine (BSA–CML), as well as BSA–methylglyoxal (BSA–MG), resulted in a significant increase in DNA damage, and this effect was mediated by RAGE [15]. Recently, we have reported on the synthetic melibiose-derived glycation product (MAGE) that mimics a unique epitope present in human and animal tissues [16]. In contrast to known classical advanced glycation end-products, the MAGEs are synthesized in anhydrous conditions. The generated mouse anti-MAGE monoclonal antibody recognized the native analogous product in living organisms. The MAGE cross-reactive autoantibodies were detected in patients with diabetes [16]. 

In this work, we present data on the genotoxic activity of different forms of MAGE, i.e., compounds obtained after protein or amino acid modification with melibiose. The genotoxic properties of the high molecular mass– and low molecular mass–MAGE (HMW–MAGE and LMW–MAGE, respectively) were determined by assessing their ability to induce micronuclei (MN) in cultured peripheral blood lymphocyte cells (PBLs) and in four human cell lines. Three cancer cell lines were of lung adenocarcinoma (A549), human melanoma (SK-MEL) and colorectal cancer line of cells with p53 off (HCT 116 (-/-)), whereas the control normal cells were BEAS-2B line from bronchial epithelium. Pro-apoptotic activity of p53 protein is associated with its anticancer effects; therefore, it was interesting to include these cells to our experiments. The effect of selected fractions of MAGE on PBLs damage after X-ray irradiation of cells has been also determined.

## 2. Materials and Methods

### 2.1. Materials 

All chemicals were purchased from Sigma–Aldrich (Poznań, Poland). Irradiation experiments were performed using X-rays (Clinac 600, 6 MV, Varian) with dose of 2 Gy (1 Gy/min). Monomeric bovine serum albumin (BSA) was prepared from Cohn fraction V BSA suitable for radioimmunoassay (Sigma–Aldrich), as described [17]. 

### 2.2. Preparation and Fractionation of HMW–MAGE and LMW–MAGE

The glycation products were obtained using dry conditions in high-temperature (HTG) using an oven or microwave and the aqueous conventional glycation conditions (ACG) in solution, as described [16,17,18]. Briefly, BSA and melibiose were dissolved in water at a ratio of 2:3 (*w*/*w*). In some experiments, myoglobin from equine skeletal muscle (MB) was used instead of BSA to obtain MB–mel products. The sample was frozen at –70 °C, lyophilized and then heated at 116–120 °C for 30 min or in microwave oven set up at 85°C for 45 min. After cooling, the material was dissolved in water, and after centrifugation, the supernatant was fractionated on a gel filtration column of Sephadex G-200 (1.6 **·** 100 cm) equilibrated with PBS containing 0.02 % (*w*/*v*) NaN_3_. Next, the material was dialyzed against distilled water to remove the unreacted material and lyophilized. BSA modified by methylglyoxal was obtained by the incubation of BSA water solution (25 mg/mL) with methylglyoxal (3.6 mg/mL) at 37 °C for 7 days [19]. Two groups of low molecular mass products were also synthesized: one from D-melibiose and *N*-α-acetyl-L-lysine (LM), and the second one from a mixture of D-melibiose, *N*-α-acetyl-L-lysine and *N*-α-acetyl-L-arginine (LMA). Substrates were dissolved in water and, after lyophilization, were heated at 116–120 °C for 30 min. The products were fractionated on HW40S gel filtration column (1.6 **·** 100 cm) in 0.1 M ammonium acetate as an eluent, and the obtained fractions were desalted on Bio-Gel P-2 column. Fractions from gel filtration columns were monitored for their activity of the inhibition of HMW–MAGE/anti-MAGE reactivity [16] and the exemplified pattern of fractionation is shown on Figure 1. The fractions active in the inhibition assay LM3A, LM3B, LM3C and LMA5A (further called fractions) were subjected to the genotoxicity experiments. 

### 2.3. Cell Culture and Treatment with MAGE

This study was performed on cultures of human peripheral blood lymphocytes obtained from four healthy donors. Samples of blood were collected to 0.015% solution of EDTA, and cultures were prepared by adding 0.5 mL of blood to 4.5 mL of Dulbecco’s Modified Eagle’s Medium supplemented with 15 % fetal calf serum and antibiotics (penicillin, streptomycin). Volumes of 1mL were dispensed to 24-well cell culture polystyrene plates (Linbro). MAGE was dissolved in medium and added to the cultures 15 min before mitotic stimulation. They were tested at a final concentration of 10–100 μg/mL. Samples were cultured in a cell culture incubator (at 37 °C in the presence of 5% CO_2_). Human peripheral blood cells treated with PBS served as controls. Experiments have been also performed on human cell lines: control normal cells BEAS-2B (bronchial epithelium) and three cancer cells A549 (lung adenocarcinoma), HCT 116 (-/-) (colorectal cancer with p53 off) and SK-MEL (human melanoma). These cells were grown in a monolayer on the plastic dishes containing DMEM/F12 medium supplemented with 10% fetal bovine serum (Immuniq) and antibiotics.

### 2.4. Micronucleus Frequency Assay

The micronucleus test on lymphocytes was performed according to Fenech and Morley procedure [20]. Lymphocytes cultured at 37 °C in humidified atmosphere containing 5% CO_2_ were stimulated for mitosis with 5 μg/mL phytohemagglutinin (Lectin). Cytochalasin B (6 μg/mL) was added 44 h later to accumulate the cells that had divided once. Following 72 h of cultivation, cells were transferred onto glass slides by cytospin–centrifugation and fixed with ethanol: acetic acid (3:1 *v*/*v*). The staining was performed by May–Giemsa dyes. Negative control samples contained PBL cultures without any treatment. Positive control sample was prepared with 10 μg/mL of bleomycin. Micronuclei (MN) were scored in 500 binucleated lymphocytes (BNL) for each experimental point. Apoptosis was scored in 1000 cells. After irradiation of cancer and normal cells, the cytochalasin B was added to the cultures to a final concentration 2 µg/mL, and cells were incubated for 48h prior to fixation. The cells were fixed in situ with a cold solution of 1% glutaraldehyde (Sigma) in a phosphate buffer (pH 7.5) and stained by Feulgen reaction. At least 500 binucleated cells were examined for the presence of micronuclei under microscope. The fraction of cells showing condensation of chromatin characteristic for the apoptosis process was also recorded. Nuclear division index (NDI) was calculated using the formula: % NDI = (1N + (2 × 2N) + (3 x 3N) + (4 × 4N))/400 cells, where: 1N is amount cells with 1 nucleus; 2N, with two nuclei; 3N, with three nuclei; and 4N, with four nuclei. Experiments were repeated on the samples of blood obtained from three donors, and results were summarized. The data are expressed as number of MN per 1000 BNL, as well as the frequency of binucleated cells containing one or more MN. 

### 2.5. Irradiation Experiments

X-ray irradiation was carried out at room temperature with a Clinac 600 GMV Machine (Varian) using a 2 Gy dose delivered at 1 Gy/min dose rate. All experiments were repeated three times.

### 2.6. Statistical Analysis

Normal distribution was analyzed by the Shapiro–Wilk test, and values are presented as mean and standard deviation. The t-Walsh test was used to analyze the statistical significance of the differences between the negative control and the test groups. Positive values of the Welch test indicate that the mean values of the test group are greater than the mean values of the control group. Negative values show the opposite relation, i.e., the mean values of the test group are lower than the mean values of the control group. The *p*-value < 0.05 was considered significant.

## 3. Results and Discussion

In order to study the effect on cells of new glycation products synthesized under anhydrous conditions (MAGE), we have applied a micronucleus (MN) frequency assay. These MAGE products mimic a unique epitope present in human tissues [16]. Increased expression of the studied epitope was observed in tissues of diabetic patients. Thus, it was important to check the genotoxic properties of synthetic model analogues of this tissue epitope. The lower resistance of diabetic patients to many common infections is due to increased cell death associated with hyperglycemia, with greater apoptosis of lymphocytes and neutrophils [21]. Loss of these cells might hinder both innate and adaptive responses to any infection [22,23]. Micronucleus frequency assay presents a valuable approach to the evaluation of human PBLs genomic damage. The MNs as DNA-containing structures are formed during mitosis and result from chromosomal breaks or from whole chromosomes incorrectly distributed during mitosis. The MN frequency test is based on using cytochalasine B in a culture of mitotically stimulated lymphocytes. This compound induces the inhibition of cytoplasm division (cytokinesis) without the effects on cell nucleus division (kariokinesis). After the completion of division process, the binuclear lymphocytes are formed, which very often contain the additional micronuclei, deriving from acentric fragments of chromosomes. This assay allows for the detection of unrepaired double strand breaks as well as damage of spindle poles during mitosis. In our experiments, the round and oval objects were classified as MN if they appeared as separated from nucleus and showed staining similar to those of nuclei, with an area less than one quarter of the area of the average normal nucleus [24]. 

In this paper, we aimed to study an effect of the in vitro formed model MAGE, namely HMW– and LMW–MAGE compounds on cells, using the micronucleus genotoxicity assay. Four fractions of HMW–MAGE formed from BSA and melibiose were selected after HTG products separation on Sephadex G-200 [16]. SDS–PAGE analysis of gel stained with Coomassie Brilliant Blue showed in fraction II presence of the soluble cross-linked MAGE products and the lowest amount of glycated protein monomer. This fraction (BSA–mel-II) was further analyzed for genotoxic activities. We also used in our experiments the other MAGE form, namely MB–mel, which displays a common antigenic structure independent of the carrier protein [16]. It was glycated under HTG anhydrous conditions by heating the lyophilized mixture of MB and mel. In addition, we tested activity of other MAGEs formed from BSA and methylglyoxal, which results in the cross-linked glycation products (BSA–MG). The whole glycation mixture after reaction was analyzed for the genotoxic activity.

In order to test the low molecular mass LMW–MAGE, the HTG reaction was carried on a lyophilized mixture of *N*-α-acetyl-L-lysine, *N*-α-acetyl-L-arginine and D-melibiose to obtain products called LMA or a mixture of *N*-α-acetyl-L-lysine and D-melibiose to obtain the LM products. The postreaction material was submitted to a gel filtration on the HW40S (several fractions collected based on the molecular mass) and Bio-Gel P-2 columns (desalting of the individual fractions). The resulting fractions were monitored with ELISA for the inhibition activity of the reaction HMW–MAGE/anti-MAGE. The representative elution profiles and respective fractions are shown on Figure 1 (see also Ref. [16]). The products active in inhibition assay (called further LMW–MAGE fractions LMA and LM) were subjected to the genotoxic analysis.

Human PBLs have been often applied as biomarkers of early effects of genotoxic compounds [24]. Our studies allowed for investigation of the HMW–MAGE effect on PBLs DNA damage and quantification of binucleated cells formed after treatment with 100 μg/mL of BSA–mel-II. The experiments showed a statistically significant increase of genomic damage in human PBL in the presence of HMW–MAGE, i.e., BSA–mel or BSA–MG fraction was concentration-dependent (Figure 2A). Micronuclei frequencies after incubation of PBLs with BSA–mel-II averaged in 40 ± 5.34/1000 CBL and was about 45% elevated, in comparison to the negative control sample (cells without any treatment). Genotoxic activity of this fraction at 100 μg/mL was markedly higher than that for BSA–MG glycation product, a reference inducer of genotoxicity (Figure 2A). We suggest that synthesized BSA–mel and BSA–MG derivatives can interact with RAGE receptors on the lymphocytes surface, followed by a downstream signaling. Stimulation of NF-κB (nuclear factor kappa-B) activates NADPH oxidase, which effectively increases the ROS (reactive oxygen species) level [10,13]. Various degrees of PBL DNA damage in our experiments are probably due to certain differences in both HMW–MAGE fractions affinities to the receptor. The higher genotoxic activity of BSA–mel-II, compared to BSA–MG products, indicates that both types of HMW–MAGE possess different epitopes. This is supported by our results [16] on the structure of MAGE epitope derived from melibiose that is distinct from structures formed from methylglyoxal [6,19].

Interestingly, treatment of cells with the synthetic low molecular mass counterparts of MAGE (LMA 5A) at a concentration range of 10–100 μg/mL did not induce micronuclei formation in the culture of human lymphocytes (Figure 2B). The low sensitivity of PBLs to LMW products may indicate the lack of their interaction with RAGE on the cell surface. Such compounds may be safe for cells after their appearance in circulation as products formed at early stages of glycation or as a result of degradation of the higher glycation adducts. 

Genotoxic activity of HMW–MAGE (Series A) and LMW–MAGEs (Series B) is shown in Table 1. The presented results concern an average from calculations made on 4000 binucleated lymphocytes. It should be mentioned that LMA products had no effect on DNA in human PBLs. Three-fold increase in number of damaged cells in the presence of bleomycin (positive control) was observed (Table 1), in contrast to the LMW–MAGEs that were not genotoxic. The BSA–mel expressed distinct genotoxic activity, compared to weaker, though also statistically significant, genotoxic BSA–MG products. 

Next, we tested whether a presence of the LMW–MAGE affected genotoxicity exerted by different HMW products, i.e., BSA–mel-II and BSA–MG (Table 2). The degree of PBLs damage was not changed in cells exposed to the BSA–mel-II nor BSA–MG, in addition to the selected LMW–MAGEs as LM or LMA products, at equivalent concentrations. Those results confirmed differences in affinity of investigated ligands to RAGE of human PBLs.

In another experiment with exposition of PBLs to 2.0 Gy X-radiation, an over 11-times increase in a frequency of the micronuclei number occurred in the analyzed 1000 binucleated cells (Table 3). Various concentrations of D-melibiose or *N*-α-acetyl-L-lysine as well as *N*-α-acetyl-L-arginine in control experiments had no effect on DNA of irradiated human PBLs. However, we observed some protection of cells against X-ray induced genotoxicity in presence of the LMW–MAGE derived from LMA products. Protective effect was concentration-dependent, and fraction of damaged cells decreased from 3 and 7 % for 1μg/mL to 15 and 17 % at 100 μg/mL level for LMA 5A and LMA 5B, respectively. The mechanism of observed effects will be investigated in the next step of our research. 

Results shown in Table 4, Table 5, Table 6 and Table 7 revealed that advanced glycation products synthesized in water conditions (ACG) did not present genotoxic activity of micronuclei formation, nor did it induce an apoptosis in studied cell lines. Glycation products synthesized in anhydrous conditions (HTG) showed genotoxic properties, dose dependent, because the number of micronuclei increased twice in cells BEAS-2B (Table 4) and A549 (Table 5), and a 3-fold increase was observed in HCT 116 (-/-) cells (Table 6). HTG compounds induced apoptosis only in BEAS-2B and A549 cells. The formation of micronuclei is p53-independent, since the same effect was noted in the HCT 116 (-/-) cells. Human melanoma cancer SK-MEL cells, like all previous studied cell lines, are sensitive to genotoxic effect of HMW–MAGE synthesized in anhydrous conditions in microwave reactor—MWG (Table 7). The LMW–MAGE products are devoid of this genotoxic activity but, instead, inhibit the activity of HMW–MAGE. These compounds have no effect on cell division index NDI. It is also of note that the HMW–MAGE induce apoptosis, while LMW–MAGE inhibit this effect.

In BEAS-2B cells, genotoxicity induced by HTG products was inhibited by different LM fractions that also inhibit apoptosis, and the results are summarized in Table 8, Table 9 and Table 10. Similarly, in A459 cells, the genotoxicity and apoptosis were also inhibited by LM fractions. Regarding HCT 116 (-/-) cells, genotoxicity is inhibited by fraction 3B, slightly weaker than by 3C and at least 3A, and weak apoptosis induction is inhibited by fraction 3B. The most active in inhibiting the genotoxic effect of MAGE synthesized with HTG method fractions will be further studied.

Our observations clearly indicate an inhibition of HTG products-induced cell genotoxicity by low molecular mass MAGE compounds and prompt us to further studies for a potential drug against the genotoxic effect of HTG products. We also find interesting the beneficial apoptotic effect of HMW–MAGE on cancer cells. Recent studies indicate that protein aggregation occurs as a result of AGE formation [25]. The cross talk between the aggregation of protein initiated with glycation may eventually lead to the development of cancer due to a promotion of the transformation of healthy cells to neoplasia, leading to tumorigenesis [26]. 

## 4. Conclusions

The new high molecular mass MAGE synthesized at anhydrous conditions from BSA and MB glycation with D-melibiose (BSA–mel-II, MB–mel) demonstrated significant genotoxic activity against human peripheral blood lymphocytes and cell lines. The MAGE derivative from melibiose expressed higher genotoxic activity than the structure formed from methylglyoxal. In contrast, the low molecular mass glycation counterparts (LMW–MAGEs) obtained after the reaction of melibiose with free amino acids were not genotoxic. The presence of the LMW–MAGE had no effect on the genotoxicity exerted by different HMW products. The presence of LMW–MAGE in circulation may protect cells against damage induced by X-irradiation. Differences in genotoxicity of investigated compounds, namely BSA–MAGE and BSA–MG, suggest various mechanisms of their interaction with cell surface. Probably some epitopes in their structure determine the affinity for receptors RAGE. Interestingly, the HMW–MAGEs induce apoptosis, while LMW–MAGEs inhibit this effect. HMW–MAGE products exerted genotoxic and apoptotic effects in melanoma and lung cancer cells and also in bronchial control cells. However, the most intense genotoxic effect was in colorectal cancer cells. The formation of micronuclei is p53-independent. Further investigations will be performed to determine potential beneficial pro-apoptotic effect on cancer cells. Advanced glycation products synthesized in water conditions (ACG) did not present genotoxic activity, nor did it induce an apoptosis in studied cell lines.

## Figures and Tables

**Figure 1 genes-12-00729-f001:**
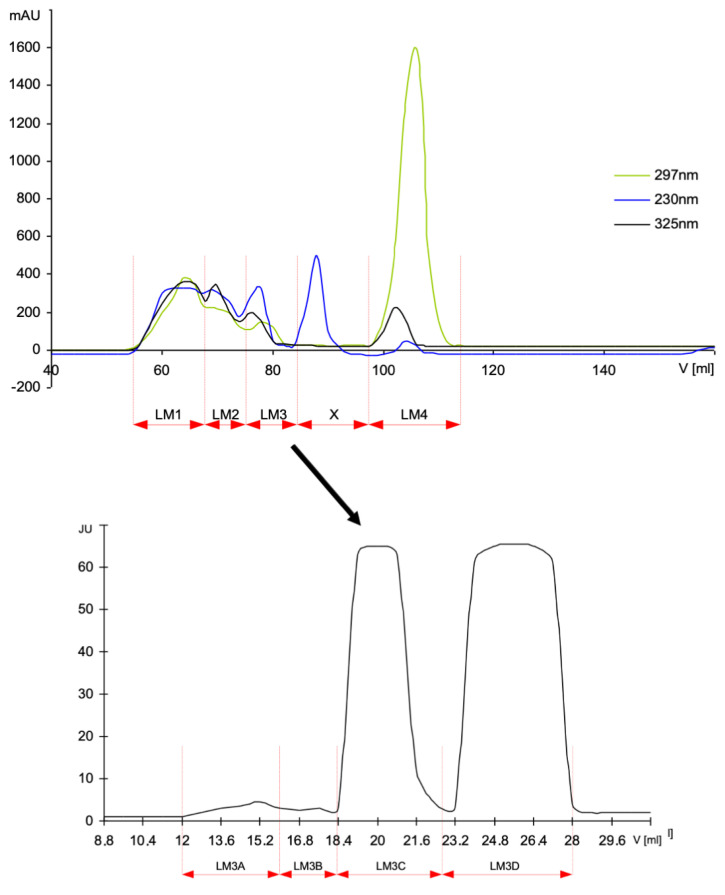
Chromatography on HW40S column in 0.1 M ammonium acetate of the LMW products formed from D-melibiose in reaction with *N*-α-acetyl-L-lysine, resulting in several fractions (upper panel), and the obtained fractions from the LM3 desalted on a Bio-Gel P-2 column (lower panel).

**Figure 2 genes-12-00729-f002:**
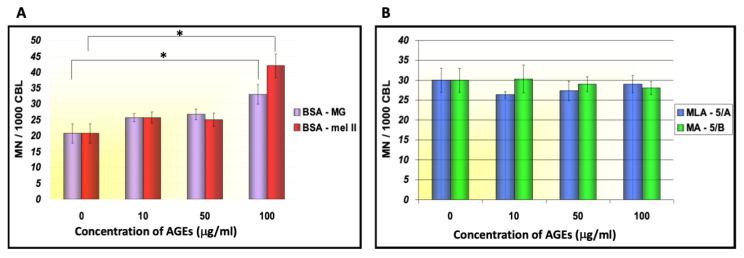
The impact of HMW–MAGE derived from BSA and D-melibiose or methylglyoxal on DNA damage in human PBL (**A**) and genotoxic activity of LMW–MAGE synthesized in reaction of D-melibiose, *N*-α-acetyl-L-lysine and *N*-α-acetyl-L-arginine (LMA) or from a mixture of D-melibiose and *N*-α-acetyl-L-lysine (LM) (**B**); * indicates *p* < 0.05.

**Table 1 genes-12-00729-t001:** Micronuclei analysis results of advanced glycation end-products genotoxicity, obtained in two sets of experiments.

Tested Compounds	Concentration (μg/mL)	Frequencies of MN/1000 CBL	*t*-Welch Test t	*p*-Value	Fraction of Damaged Cells (Promille)	t-Welch Test t	*p*-Value
Negative control	-	20.66 ± 1.39	-	-	19.3 ± 1.84	-	-
BSA monomer	100	18.00 ± 1.61	−27.964	<0.005	16.3 ± 1.52	−28.107	<0.005
Melibiose	100	18.61 ± 1.69	−20.948	<0.005	17.8 ± 1.24	−15.117	<0.005
BSA–mel fr. II HTG	100	40.22 ± 5.34	79.264	<0.005	38.0 ± 5.12	76.857	<0.005
BSA–mel fr. II HTG	50	25.70 ± 1.87	48.368	<0.005	24.6 ± 2.82	35.196	<0.005
BSA–MG	100	28.80 ± 6.50	27.383	<0.005	29.3 ± 4.09	49.858	<0.005
BSA–MG	50	24.89 ± 5.26	17.385	<0.005	21.5 ± 2.88	14.394	<0.005
LM 3C	100	17.30 ± 1.25	−40.191	<0.005	17.0 ± 1.01	−24.502	<0.005
LMA 5A	100	19.20 ± 1.63	−15.240	<0.005	17.4 ± 1.66	−17.144	<0.005
Negative control	-	29.25 ± 2.77	-	-	27.00 ± 0.71		
Bleomycin	10	105.50 ± 7.76	206.93	<0.005	82.50 ± 5.22	235.57	<0.005
*N*-α-acetyl-L-lysine	100	30.20 ± 2.99	5.21	<0.005	26.20 ± 1.47	−10.95	<0.005
*NN*-α-acetyl-L-arginine	100	28.46 ± 2.21	−4.98	<0.005	26.00 ± 1.51	−13.40	<0.005
Melibiose	100	27.75 ± 1.92	−9.95	<0.005	26.40 ± 2.01	−6.29	<0.005
LMA 5A	10	26.33 ± 0.94	−22.32	<0.005	24.67 ± 1.89	−25.80	<0.005
	50	27.50 ± 2.06	−11.33	<0.005	25.75 ± 2.86	−9.48	<0.005
	100	30.25 ± 1.79	6.78	<0.005	28.25 ± 2.05	12.88	<0.005
LMA 5B	10	30.33 ± 4.03	4.93	<0.005	28.30 ± 2.62	10.71	<0.005
	50	29.00 ± 2.16	−1.59	0.1118	25.33 ± 1.25	−25.98	<0.005
	100	27.50 ± 1.66	−12.11	<0.005	25.00 ± 1.87	−22.36	<0.005

BSA–mel fr. II HTG and BSA–MG: fractions of HMW–MAGE obtained after BSA glycation with D-melibiose or methylglyoxal, respectively; LM 3C, LMA 5A and LMA 5B: fractions of LMW–MAGE obtained after HW40S gel filtration of products from D-melibiose reaction with *N*-α-acetyl-L-lysine (LM 3C) or reaction products of D-melibiose with *N*-α-acetyl-L-lysine and *N*-α-acetyl-L-arginine (LMA 5A, LMA 5B); MN/1000 CBL: micronuclei occurring in binucleated lymphocytes; Negative control: culturing medium.

**Table 2 genes-12-00729-t002:** Damage of human PBLs DNA in the presence of equivalent concentrations of HMW– and LMW–MAGE in the cell culture.

Tested Compound	Concentration (μg/mL)	Frequency of MN/1000 CBL	*t*-Welch Test t	*p*-Value	Fraction of Damaged Cells (promille)	*t*-Welch Test t	*p*-Value
Negative control	-	23.30 ± 1.24	-	-	21.60 ± 0.94	-	-
BSA–mel-II	100	41.33 ± 2.86	129.33	<0.005	36.00 ± 2.94	104.32	<0.005
BSA–mel-II + LM 3C	100 100	42.66 ± 4.49	92.94	<0.005	36.10 ± 3.29	94.76	<0.005
BSA–mel-II + LMA 5A	100 100	42.00 ± 2.94	131.05	<0.005	36.33 ± 2.05	146.05	<0.005
BSA MG	100	32.33 ± 1.69	96.33	<0.005	29.66 ± 1.24	115.83	<0.005
BSA–MG + LM 3C	100 100	33.70 ± 1.69	110.94	<0.005	30.00 ± 1.41	110.84	<0.005
BSA–MG + LM 2D/2	100 100	33.86 ± 2.04	98.91	<0.005	30.10 ± 0.82	152.37	<0.005

BSA–mel-II and BSA–MG: fractions of HMW–MAGE obtained after BSA glycation with D-melibiose or methylglyoxal, respectively; LM 3C and LMA 2D/2: fractions of LMW–MAGE obtained after HW40S gel filtration of products from D-melibiose reaction with only *N*-α-acetyl-L-lysine (LM 3C) or product reaction of D-melibiose with *N*-α-acetyl-L-lysine and *N*-α-acetyl-L-arginine (LMA 2D/2); Negative control: culturing medium.

**Table 3 genes-12-00729-t003:** Effect of advanced glycation end-products on the formation of micronuclei induced by X-radiation in human lymphocytes in vitro.

Tested Compounds	Concentration (μg/mL)	Frequency of MN/1000 CBL	*t*-Welch Test t	*p*-Value	Fraction of Damaged Cells (promille)	*t*-Welch Test t	*p*-Value
Negative control	-	28.35 ± 3.49	-	-	25.50 ± 2.04	-	-
X-rays (2 Gy)	-	322.50 ± 21.61	300.47	<0.005	287.25 ± 25.58	228.08	<0.005
LMA 5A + irradiation	1	310.17 ± 12.01	503.86	<0.005	268.05 ± 30.26	178.83	<0.005
	10	297.72 ± 10.11	563.16	<0.005	241.00 ± 15.05	317.28	<0.005
	100	289.25 ± 13.63	414.64	<0.005	237.50 ± 12.22	382.63	<0.005
LMA 5B + irradiation	1	341.00 ± 11.37	587.8	<0.005	277.60 ± 12.04	461.62	<0.005
	10	298.00 ± 14.9	394	<0.005	244.36 ± 8.50	559.85	<0.005
	100	297.75 ± 10.22	557.8	<0.005	245.50 ± 8.77	546.34	<0.005
*N*-α-acetyl-L-lysine + irradiation	1	313.00 ± 21.33	294.49	<0.005	252.10 ± 15.51	323.9	<0.005
	10	337.44 ± 20.42	333.63	<0.005	266.00 ± 20.61	259.66	<0.005
	100	310.33 ± 7.89	730.84	<0.005	251.00 ± 8.48	578.12	<0.005
Melibiose + irradiation	1	319.00 ± 10.71	576.97	<0.005	249.67 ± 8.34	583.82	<0.005
	10	340.62 ± 10.66	622.51	<0.005	264.50 ± 11.94	441.19	<0.005
	100	298.30 ± 22.66	263.28	<0.005	240.67 ± 14.29	333.31	<0.005
*N*-α-acetyl-L-arginine + irradiation	1	332.00 ± 16.39	405.18	<0.005	261.10 ± 13.34	390.38	<0.005
	10	330.00 ± 19.03	348.63	<0.005	266.67 ± 15.82	338.08	<0.005
	100	341.65 ± 17.40	394.76	<0.005	262.00 ± 21.18	248.53	<0.005

Abbreviations: see Table 1.

**Table 4 genes-12-00729-t004:** The effect of advanced glycation end-products on BEAS-2B cells.

Tested Compounds	Concentration (μg/mL)	Cells with Micronuclei (%)	Number of Micronuclei in 100 Cells	Cell Division Index NDI (%)	Apoptotic Cells (%)
Negative control	-	2.9	2.9	2.1	3.3
MB–mel ACG	1	2.9	2.9	2.09	3.2
	10	2.9	2.9	2.01	3.5
	20	2.8	2.8	2.07	3.5
	50	2.8	2.9	1.99	3.6
	100	3.3	4.1	2.01	3.5
MB–mel HTG	1	3.4	3.4	2.04	3.5
	10	4	4.2	1.95	3.8
	20	5.1	5.7	2.02	3.9
	50	5.7	6	2.09	4
	100	5.7	6.5	1.99	3.9
Melibiose	100	2.8	2.9	2.1	3.4
Myoglobin	100	2.9	2.9	2.02	3
Bleomycin	10	13.5	19.5	1.33	5

**Table 5 genes-12-00729-t005:** The effect of advanced glycation end-products on A549 cells.

Tested Compounds	Concentration (μg/mL)	Cells with Micronuclei (%)	Number of Micronuclei in 100 Cells	Cell Division Index NDI (%)	Apoptotic Cells (%)
Negative control	-	2.2	2.1	2.09	1.1
MB–mel ACG	1	2.1	2.1	2.00	1.1
	10	2.2	1.6	1.99	1.2
	20	2.1	2.2	1.98	1.2
	50	2.2	2.4	2.04	1.2
	100	2.1	2.3	2.00	1.2
MB–mel HTG	1	2.3	2.7	2.00	1.4
	10	2.8	3.3	1.96	1.7
	20	3.0	3.6	2.03	1.6
	50	3.1	3.8	1.96	1.7
	100	3.3	4.3	1.95	2.4
Melibiose	100	2.1	2.1	2.05	1.1
Myoglobin	100	2.1	2.2	2.08	1.2
Bleomycin	10	15.9	24.4	1.56	2.7

**Table 6 genes-12-00729-t006:** The effect of advanced glycation end-products on HCT 116 (-/-) cells.

Tested Compounds	Concentration (μg/mL)	Cells with Micronuclei (%)	Number of Micronuclei in 100 Cells	Cell Division Index NDI (%)	Apoptotic Cells (%)
Negative control	0	1.5	1.2	1.94	1.3
MB–mel ACG	1	1.6	1.6	1.99	1.4
	10	1.5	1.6	1.99	1.3
	20	1.5	1.6	1.93	1.4
	50	1.5	1.6	2.01	1.3
	100	1.7	1.8	2.01	1.5
MB–mel HTG	1	1.8	1.8	1.99	1.3
	10	3.5	3.8	1.98	1.3
	20	4.4	5.3	1.92	1.5
	50	4.9	5.2	1.95	1.4
	100	6.2	6.8	1.94	1.6
Melibiose	100	1.5	1.5	2.01	1.3
Myoglobin	100	1.4	1.1	1.98	1.3
Bleomycin	10	11.6	15.2	1.37	2.3

**Table 7 genes-12-00729-t007:** The effect of advanced glycation end-products on SK-MEL cells.

Concentration (μg/mL)		Cells with Micronuclei (%)			Cell Division Index NDI (%)			Apoptotic Cells (%)		
MB–mel HTG	LM 3C									
0	0	1	2	1	1.92	1.97	1.84	0.8	0.6	0.5
10	0	2	2	1	2.04	1.95	1.92	0.7	0.5	0.8
20	0	4	4	5	1.92	1.94	1.86	0.9	1.2	0.9
50	0	5	4	6	2	1.92	1.94	1.5	1.7	1.4
0	10	1	1	2	1.96	2.08	2.05	0.5	0.5	0.7
0	20	2	1	1	2.08	1.9	1.92	0.4	0.6	0.6
0	50	1	2	2	1.95	1.92	1.98	0.6	0.5	0.8
10	10	2	1	1	1.87	1.95	1.94	0.5	0.7	0.7
20	20	3	2	2	1.98	2.1	1.95	0.6	0.6	0.7
50	50	3	3	2	2.04	1.96	1.94	0.5	0.7	0.5
Bleomycin 10 μg/ml		18	21	17	1.47	1.68	1.52	2.8	3	2.4

**Table 8 genes-12-00729-t008:** The effect of LMW–MAGE on genotoxic activity of HTG glycation products in BEAS-2B cells.

	Concentration (μg/mL)	Cells with Micronuclei (%)			Number of Micronuclei in 100 Cells			Apoptotic Cells (%)		
Negative control	-	3.1	2.6	2.8	3.1	2.6	2.8	3.0	3.2	3.2
MB–mel HTG	20	6.0	5.2	5.7	6.4	5.5	6.1	3.8	4.1	3.6
MB–mel HTG + LM 3A	20 10	5.0	6.1	5.5	5.6	6.3	5.7	3.5	3.9	3.3
MB–mel HTG + LM 3B	20 10	4.0	4.8	4.4	4.2	4.8	4.8	3.1	3.3	3.3
MB–mel HTG + LM 3C	20 10	4.2	4.6	4.7	4.2	4.8	4.9	3.5	3.2	3.4
Melibiose	100	2.7	3.1	3.0	2.7	3.1	3.0	3.2	3.0	3.0
Myoglobin	100	2.6	2.8	3.1	2.6	2.8	3.1	3.0	3.2	2.9
LM 3A	10	3.0			3.0			2.9		
LM 3B	10	2.8			2.8			3.1		
LM 3C	10	3.0			3.0			2.9		

**Table 9 genes-12-00729-t009:** The effect of LMW–MAGE on genotoxic activity of HTG glycation products in A549 cells.

	Concentration (μg/ ml)	Cells with Micronuclei (%)			Number of Micronuclei in 100 Cells			Apoptotic Cells (%)		
Control	0	2.0	1.9	2.2	2.0	1.9	2.2	1.0	1.2	1.0
MB–mel HTG	20	3.4	3.8	3.3	3.6	4.1	3.5	1.4	1.8	2.0
MB–mel HTG + LM 3A	20 10	3.6	3.4	3.2	3.6	3.6	3.4	1.5	1.7	2.9
MB–mel HTG + LM 3B	20 10	2.6	2.8	3.0	2.6	3.0	3.4	1.1	1.3	1.1
MB–mel HTG + LM 3C	20 10	2.7	3.1	3.0	2.7	3.3	3.0	1.2	1.0	1.3
Melibiose	100	1.8	2.0	2.0	1.8	2.0	2.0	1.1	0.9	1.0
Myoglobin	100	2.0	2.2	2.0	2.0	2.2	2.0	1.0	1.0	1.2
LM 3A	10	1.9			1.9					0.9
LM 3B	10	2.2			2.2					1.1
LM 3C	10	2.0			2.0					1.0

**Table 10 genes-12-00729-t010:** The effect of LMW–MAGE on genotoxic activity of HTG glycation products in HCT 116 (-/-) cells.

	Concentration(μg/mL)	Cells with Micronuclei(%)	Number of Micronuclei in 100 Cells	Apoptotic Cells (%)
Negative control	0	1.3	1.4	1.8	1.3	1.4	1.8	1.5	1.1	1.2
MB–mel HTG	20	4.5	4.2	4.8	5.5	5.2	5.8	1.5	1.8	1.6
MB–mel HTG+ LM 3A	2010	4.6	4.0	4.4	5.2	4.4	4.6	1.5	1.3	1.6
MB–mel HTG+ MB 3B	2010	3.8	3.6	3.5	4.0	4.2	3.8	1.4	1.1	1.4
MB–mel HTG+ LM 3C	2010	3.9	3.7	4.2	4.5	4.4	4.2	1.2	1.6	1.5
Melibiose	100	1.3	1.7	1.5	1.3	1.7	1.5	1.5	1.1	1.4
Myoglobin	100	1.6	1.6	1.8	1.6	1.6	1.8	1.0	1.4	1.2
LM 3A	10	1.4			1.4			1.2		
LM 3B	10	1.7			1.7			1.0		
LM 3C	10	1.4			1.5			1.4

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
