# Peer review of "The Genotoxic and Pro-Apoptotic Activities of Advanced Glycation End-Products (MAGE) Measured with Micronuclei Assay Are Inhibited by Their Low Molecular Mass Counterparts"

_genes, 2021, doi:10.3390/genes12050729_

Round 1

Reviewer 1 Report

Materials and Methods

Please add more information about the protocol for cells' culture.

Why do you not explain the statistical analysis?

Which test do you use for the counting data? (Mann Whitney U test, Wilcoxon test analysis, or other) Results and discussion

Page 5 line 186

In this page the authors describe their investigation of the HMW-MAGE effect on PBLs DNA damage and quantification of binucleated cells formed after treatment with 100 μg/ml of BSA-mel-II presented on Figure 2. Unfortunately, in the same page and line 205 they present one figure without control panel. Moreover, this figure is presented in high magnification.

Please add untreated lymphocytes as controls, and cells after treatment with 100 μg/ml of BSA-mel-II, at least in 2 different magnifications.

Page 6

Please add p value to figure 3.

All tables in this article are presented without statistical analysis.

Author Response

Dear Reviewer,

the answers to your questions are as follows:

The introduction, as well as research design and conclusions, have been improved in the new version of the manuscript and all changes are marked in yellow.

Regarding the description of methods and results, these chapters are improved thoroughly with Table 1 completed, due to its lacking, although the Experimental series B was submitted, statistical details are added, the sentences are shortened to be in a better style, English is corrected with errors cleared.

In the Materials and Methods part, the lacking information is added to the cells` culture.

The methodology of statistical analysis is described. The appropriate test is used for statistical analysis performed and all results are included in Tables.  

Regarding Figure 2, it is moved to the Material and Methods section because this picture is dedicated only to present the methodology, to show the shape of binucleated cells treated in a model experiment with X-rays, magnified for better presentation. The HMW-MAGE effect on PBLs DNA damage and quantification of binucleated cells formed after treatment with 100 μg/ml of BSA-mel-II is presented in the Results section, and together with all performed controls are included in Tables. Therefore it is not necessary to add untreated lymphocyte picture and cells after treatment with 100 μg/ml of BSA-mel-II, in different magnifications.

Figure 3 is completed for p-value, as well as results of statistical analysis are added to tables where applicable.

Thank you very much for your kind help in improving the manuscript.

Reviewer 2 Report

I strongly suggest that the authors split those super long sentences (such as paper’s title; line 18-21; line 21-24; line 41-44; line 48-50; line 73-79; line 96-100…) in their manuscript.

Grammar errors such as line 29 (“DNA damage the cells”); line 82 (“have” should be replaced with “has”); Line 189 (“on Fig. 2” should be corrected to “in Fig. 2”), etc.

Replace RIA with radioimmunoassay (line 88).

Author Response

Dear Reviewer,

the answers to your questions are as follows:

Concerning the English language style, it is improved in the new version of the manuscript.

Presentation of results is improved for better clarity.

Too long sentences are divided for shorter expressions in the whole text of the manuscript in the new revised version. Also, grammar errors are corrected in the manuscript. RIA abbreviation is explained and written as radioimmunoassay.

All changes are marked in yellow in the new version.

Regarding other improvements, additions, and changes, the note the editors are as follows:

  1. the title is shortened for better clarity,
  2. the name of one author who deceased, dr. Z. Maniakowski is marked with a frame
  3. one author is added who participated in studies, namely dr. Łukasz Łaczmański
  4. name of the Institute of Oncology in Gliwice is corrected as well as Laboratory of prof. M. Staniszewska
  5. references part is corrected

Thank you very much for your help in improving the manuscript.

Round 2

Reviewer 1 Report

References

All references must change according to journal’s instruction.

Materials and Methods

Please detailed all process of culturing human peripheral blood lymphocytes from whole blood.

Why do you not explain the statistical analysis?

Results and discussion

Page 4: the authors describe their investigation of the HMW-MAGE effect on PBLs DNA damage and quantification of binucleated cells formed after treatment with 100 μg/ml of BSA-mel-II presented on Figure 2. Unfortunately, this figure is presented without control panel. Please add untreated lymphocytes as controls, and cells after treatment with 100 μg/ml of BSA-mel-II, at least in 2 different magnifications.

Author Response

Dear Editors,

Regarding the Reviewers questions to the second version of manuscript, the third version is further improved and the answers to these questions are as follows:

Reviewer 1

The research design, description of methods and presentation of results have been further improved in new version of manuscript. Regarding the conclusions they are further completed to reflect the results obtained. All changes are marked in blue in the current version of manuscript.

References have been adjusted to the journal instructions.

In Materials and Methods section the details of culturing of human peripheral blood lymphocytes from whole blood are added to the text of new version of manuscript.

Statistical analysis is explained in some details according to reviewer advice.

In Results and Discussion section on Page 4 the proposed Figure 2 was a selected picture of binucleated lymphocyte obtained with X-ray irradiation. It was representative image of a binucleated human lymphocyte incubated for 3 days after irradiation with 2 Gy of X-ray, presented in 400 magnification for better visualization. It was presented to show the effect of X-ray irradiation on cells and is a methodologic detail showing how this method works. This picture was not linked with treatment cells with 100 μg/ml of BSA-mel-II. The Figure 2 was erroneously placed with BSA-mel treatment, another experiment. This error was improved by placing the Figure 2 to Methods section in second version of manuscript. However, in order to have better clarity of text we remove the Figure 2 from the revised version, because it has no influence on results, as is no necessary. The method is well known (Fenech and Morley procedure [20]), routinely used in laboratories and described in details previously, so it is not necessary to present the cell after X-ray irradiation, this is not a goal of the paper.   

We thank to reviewers for help in further improving the manuscript that we believe it has a biomedical importance. The biological activities of studied compounds are related to newly discovered epitope in human and animal tissues. We hope the revised version of manuscript will be acceptable for publishing, ready for further processing.

Reviewer 2 Report

All my concerns are properly addressed in the new version. I do not have any further comments.

Author Response

Dear Editors,

Regarding the Reviewers questions to the second version of manuscript, the third version is further improved and the answers to these questions are as follows:

Reviewer 2

Thank you very much for the acceptance of the manuscript in that form and your help in its improving. The style has been further improved in revised version.

We thank to reviewers for help in further improving the manuscript that we believe it has a biomedical importance. The biological activities of studied compounds are related to newly discovered epitope in human and animal tissues. We hope the revised version of manuscript will be acceptable for publishing, ready for further processing.